# Optimizing Public Charging: An Integrated Approach Based on GIS and Multi-Criteria Decision Analysis

**Ali Khalife, Tu-Anh Fay \*** 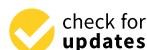 **and Dietmar Göhlich**

Methods for Product Development and Mechatronics, Technische Universität Berlin, 10623 Berlin, Germany;
ali.z.khalifeh@gmail.com (A.K.); dietmar.goehlich@tu-berlin.de (D.G.)
**\*** Correspondence: tu-anh.fay@tu-berlin.de

**Abstract:** The rise in electric vehicle uptake has reshaped the German mobility landscape at unprecedented speed and scale. While public charging is pivotal to growing the electric vehicle market, municipalities can play a crucial role in accelerating the energy transition in transport. This research aims to assist municipalities in planning their strategic rollouts of public charging infrastructure in size and location. In the first step, charging demand is estimated based on four development scenarios in 2030 of EV adoption and public charging. In a second step, a geospatial analysis was performed on the study area. Supply and demand criteria were considered to reflect the attractiveness of each location on a grid map. While the supply criteria represent constraints related to infrastructure availability, the demand criteria are categorized into three dimensions: residential, commercial, and leisure. The prioritization of demand criteria was derived from the municipality's input using the analytical hierarchy process method to reflect its strategy. After obtaining the suitability index map, a cluster analysis was performed using a k-means clustering algorithm to ensure adequate geographical coverage of the charging network. Finally, the proposed charging stations in each scenario were allocated to the top-scoring locations, establishing a municipal public charging network.

**Keywords:** electric vehicles; public charging infrastructure planning; municipal planning; charging demand; multi-criteria decision making (MCDM); analytical hierarchy process (AHP); geospatial analysis; geographic information systems (GIS)

## 1. Introduction

The emergence of electric vehicles as climate-friendly mobility has stimulated many governments to introduce supportive policies to boost EV uptake and charging infrastructure deployment. For example, with the launch of the Climate Action Program, the German government has set the ambitious goal of having up to 10 million EVs and 1 million charging stations by 2030 [1]. Following the significant allocated funds, the mobility scene is set for a progressive change towards electrification. Several studies show that a substantial transition towards zero emission mobility systems is feasible [2]. However, to reach this goal an optimized roll-out of public chargers in cities and highway networks is required. In addition, closing the charging gap will potentially communicate to the public the maturity and reliability of EV technology. This will eventually break down the range anxiety barrier, accelerating consumers' acceptance of EVs and influencing their purchasing decisions.

Many public authorities and companies have acknowledged the importance of developing targeted public charging infrastructure. However, many uncertainties revolve around determining the necessary supply of charging stations and their spatial distribution [3]. The acceleration in EV adoption has generated different demand levels that are hard to anticipate within cities. An undersized public charging network will discourage people from adopting EVs and eventually fail to reap its benefits. On the other hand, an oversized network will draw higher levels of investment and yield to unrecoverable losses. The distribution of charging stations that are commonly located in high-traffic zones limits

the effectiveness of these stations and disregards the location and users' properties. In summary, there is a need to better assess public charging demand through identifying the influencing factors and following an inclusive approach to locate charging stations. Therefore, the following research question is addressed in this research:

How should the public charging network develop in order to meet potential demand and infrastructure readiness feasibly?

Consequently, this work aims to provide a holistic approach to support municipalities in developing their charging infrastructure effectively and to serve as a benchmark for responding to future demand. Based on quantitative and geospatial analysis of empirical data, it determines the volume and locations of the required charging stations.

## 2. State of the Art

An extensive literature on finding the optimal location of EV charging stations has considerably been examined by many empirical studies. While part of the research work follows a network optimization algorithm approach, other studies follow geospatial and statistical approaches. The level of the study area ranges from a country level down to the precise placement of charging stations on a street level.

In a broad systemic review related to EV and charging stations conducted by Pagany et al. [4], 119 publications followed a spatial localization modeling approach. From this set of 119, the review investigates 61 empirical studies applied to real case studies and is proven to be scalable. Moreover, it was found that a limited number of studies utilized geospatial analysis for locating charging stations. Overall, the studied models were divided into three orientations: users, destinations, or routes. Models based on users rely on demographic data, such as population, age, income, and their geo-referenced distribution. The general location planning approach is to identify potential EV adopters in a predefined area based on the mentioned attributes [5–8]. Other models concentrate on user destinations which are also referred to predefined points of interest (POI). They study the type of destinations, frequency of visitors, and dwelling times to determine the hotspots for potential charging demands [9,10]. Many spatial and temporal studies intend to support electrical grid operators in detecting possible overloads within the grid. EV adoption development and charging station location and utilization is often out of their scope. For instance, Straub et al. estimate the charging demand based on mobility profiles and vehicle specifications at a district level assuming full private fleet electrification while only considering home charging [11]. The third group focuses on travel routes and employs optimization methods. The road network is divided into nodes, where traffic is simulated through origin–destination matrices by real or synthetic travel data [12,13]. To a large extent, the overarching objective of most of the reviewed models is to cater to the charging demand with a minimal number of charging stations. The output results of the different identified models in this review take the form of demand heatmaps, partitions, or network layouts that explicitly locate the charging stations within the study area.

Another approach by Pallonetto et al. was based on analyzing the existing Irish charging network data [14]. The researchers clustered charging points with similar daily usage profiles for different charger types and locations. A congestion metric was derived and evaluated for each cluster. Their results highlight that areas with high congestion and usage require immediate expansion compared to areas with high usage and lower congestion. Y. Yang et al. evaluated the existing charging network by combining it with an electric taxi trajectory dataset on a city-level [15]. The method aimed to remove redundant charging stations in low-congestion areas while identifying those in high-congestion areas.

Similar to the approach followed in this paper, few studies applied geospatial analysis for locating charging stations. In his case study, Andrenacci et al. applied the k-means clustering method to locate the charging stations at a minimal distance to the areas of high charging demand, relying on passenger vehicle travel data [16]. Alternatively, Namdeo et al. developed a multidimensional geospatial model that incorporates socio-economic factors [7]. The outcome is a spatial plot that presents the weighted sum of the statistical layers

and the suggested public charging locations. Comparable approaches were also adopted in several studies based on similar criteria selection [10,17,18]. However, and common to all, infrastructure supply—such as parking spaces and power grid connection—were not considered.

In a more closely related study, Guler and Yomralioglu proposed an approach that integrates GIS techniques and multi-criteria decision methods to determine the optimal locations of charging stations [19]. Notably, the study combined criteria from the demand and supply sides. Nevertheless, the population density indicator did not exclusively represent the potential age group of vehicle owners. In addition, their study did not incorporate the electricity network, which has a significant impact on the location suitability from a technical and economic perspective. Distinctively, Erbaş et al. considered the proximity to electrical substations as one of their criteria selection [20]. In their future outlook, the authors highlighted that the use of technical criteria—such as power lines— offer a major improvement to their approach.

Coupled with the location model, it is vital to estimate the required number of public charging stations. Empirical studies tend to model the charging demand based on mobility survey data, predetermined charging strategies, and defined EV adoption scenarios. Several studies consider micro-simulated activity-based models (AB models), which are behavioral models used to predict activity schedules to analyze the spatial and temporal patterns of EV use and charging behavior [6,21]. Philip and Wiederer presented a different demand model using an energy balance formula [22]. In simple terms, a share of the energy consumed by EVs should be supplied by the public charging stations. Consequently, the formula allows the calculation of the public charging stations quota to electric cars based on several parameters.

To build on the previous studies outlined above, and overcome their shortcomings, this paper pursues an integrated data-driven approach with a multitude of enhancements. The start point of this paper is to determine the public charging demand. By considering the charging network, a public investment venture, the charging demand model developed by Philip and Wiederer was applied with adaptations to capture local characteristics such as passenger EV stock evolution, driving behavior, and preferential selection of charging stations type.

Several considerations were taken into account to find optimal locations of charging stations. By using a higher spatial resolution, this paper provides a better representation of the statistical data to identify potential public charging demand zones. The criteria selected and processed incorporate a different set of criteria such as proximity to the electricity grid, traffic flow, and types of buildings. Moreover, an improved representation of potential EV adopters was determined through household and vehicle registration segmentation. Furthermore, to ensure locality, the decision-makers' inputs were considered to complement this research.

## 3. Methodology

To support effective development on a municipal level, this study investigates the public charging demand and the infrastructural supply empirically and geo-spatially to determine the optimal volume and locations of public charging stations. The presented modeling approach is then applied to the city of Bottrop in North Rhine-Westphalia, Germany, as a case study. As illustrated in Figure 1, the methodology consists of three primary steps: estimation, location, and optimization.

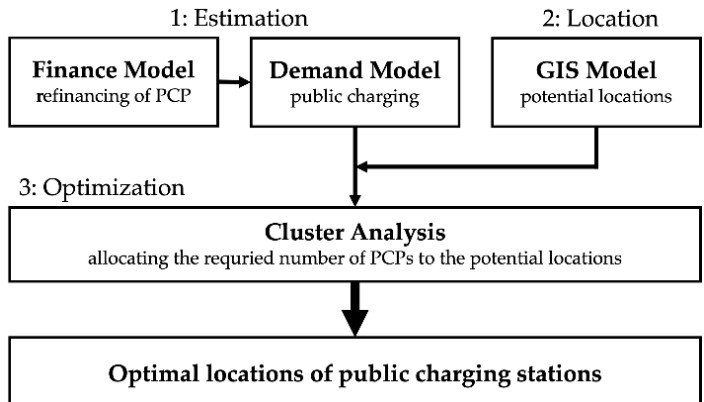

**Figure 1.** Overview of methodological approach.

### 3.1. Demand and Finance Model

In order to estimate the required number of public charging stations, an approach proposed by Philip and Wiederer [22] and extended by Wirges et al. [8] was adopted in this part. This approach is built upon an energy balance formula combined with a finance model for refinancing public charging stations, as shown in Figure 2.

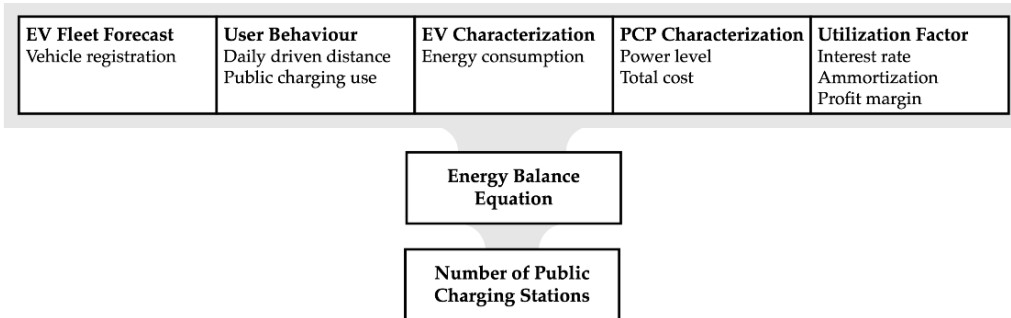

**Figure 2.** Demand and finance model.

The goal is to derive the optimal and profitable number of public charging stations. In principle, the formula attempts to balance the amount of energy consumed by EVs to the amount of energy supplied from public charging stations. The selection was in part guided by the ability of the formula to address the public charging demand exclusively while considering its economic feasibility. Contrary to the traffic simulation method, the formula relies on open government data along with flexible assumptions and is therefore more practical and adaptable. In a given time, the energy balance formula is defined as:

$$N \cdot e \cdot d \cdot R_i = C_i \cdot p_i \cdot 24 \ [\text{h}] \cdot U_i \tag{1a}$$

$i$    type of public charging station (CS) (normal, fast)
$N$   number of electric vehicles
$e$    energy consumption of an electric vehicle (kWh/km)
$d$    daily driven distance (km)
$R_i$   percentage of consumed energy recharged at public CS of type $i$ (%)
$C_i$   number of public CS of type $i$
$p_i$   power of public CS of type $i$ (kW)
$U_i$   utilization rate of public CS of type $i$ (%)

From the left side, the product $N \cdot e \cdot d$ represents the total energy consumed by EVs daily. To determine the total daily energy consumed by EVs from public charging, this product is multiplied by the percentage of public recharging $R_i$. On the right side, the product $C_i \cdot p_i \cdot 24 \ [\text{h}]$ corresponds to the total energy that could be supplied by all the public

charging stations each day. Multiplying this by the utilization factor $U_i$ yields to the actual energy withdrawn from public charging stations per day. By rearranging the formula, a straightforward calculation of the required number of charging stations is obtained, based on established parameter settings:

$$C_i = \frac{N \cdot e \cdot d \cdot R_i}{p_i \cdot 24 \ [\text{h}] \cdot U_i} \tag{1b}$$

The utilization of public charging stations, denoted by $U_i$, depends essentially on the economic model as shown below. For a charging station to be economically feasible, a minimum usage should be maintained to amortize its capital and operational costs over a period of time. In their study, Wirges et al. defined the level of utilization of charging stations as

$$U_i = \frac{t_d}{24 \ [\text{h/d}]} \tag{2}$$

where $t_d$ [h/d] is the daily total charging time in hours of a charging station.

In principle, the expected yearly profit generated from the charging station should uniformly pay back the investment cost (including interest) within a defined period. Excluding the tax deductions related to depreciation, the ordinary annuity formula from finance is utilized for determining the expected annual profit $A$ [€/year] over a defined period:

$$A = c \cdot \frac{(1+i)^{t_a} \cdot i}{(1+i)^{t_a} - 1} \tag{3a}$$

$c$     capital investment, in this case the total capital investment (€)
$i$      annual interest rate, cost of investment loan or opportunity cost of capital (%)
$t_a$    period of amortization (years)

However, as it is with public investment decisions, they are not always driven by economic profitability, but rather political objectives that pursue national interest and social welfare. In this case, the public investment in charging stations might be financed through public funds and is not evaluated based on opportunity cost. Therefore, the ideal case would be to invest in a charging station that can generate profit only to repay itself over an amortization period equals to its lifetime. In this case:

$$A = \frac{c}{t_a} \tag{3b}$$

Equivalently, the pricing model is assumed to be pay-per-use. This means that the generated daily profit of each charging station is a marginal profit $m$ (€/kWh) between charging price and electricity cost over the daily charging time $t_d$. Therefore, the yearly profit $P_y$ generated per charging station could be calculated as a function of output power, marginal profit, and average daily charging time as shown below:

$$P_y = t_d \cdot m \cdot p_i \cdot 365 \ [\text{d/y}] \tag{4}$$

Hence, to achieve economic feasibility, it is required that the actual generated yearly profit equals or exceeds the expected yearly profit presented above:

$$P_y \geq A \tag{5}$$

This leads to

$$t_d \cdot m \cdot p_i \cdot 365 \geq \frac{c}{t_a}$$

To calculate the minimum average charging time $t_d$, the formula can be rearranged to

$$t_d \geq \frac{c}{m \cdot p_i \cdot 365 \cdot t_a} \tag{6}$$

Finally, by solving for $t_d$ in Equation (2), the minimum utilization rate $U_i$ is therefore

$$U_i \geq \frac{c}{m \cdot p_i \cdot 365\,[\mathrm{d/y}] \cdot 24[\mathrm{h/d}] \cdot t_a} \tag{7}$$

### 3.2. GIS Model

The geospatial analysis was conducted on QGIS which is an open-source geographic information system (GIS) software used for processing and visualizing geodata [23]. In this study, part of the vector-based datasets were provided by the municipality of Bottrop, while the other part was obtained from OpenStreetMap [24] which is an open-source database. Further details to the used dataset are explained in the Section 4 case study. Figure 3 presents all steps performed in the geospatial analysis.

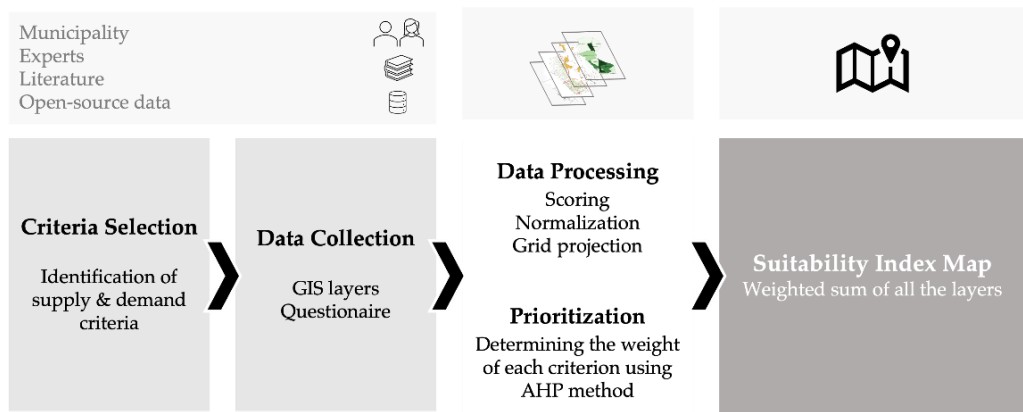

**Figure 3.** GIS model.

In order to identify and select the evaluation criteria for optimal locations of EV charging stations the literature has been reviewed and experts have been interviewed. The criteria are then classified into supply and demand criteria. Supply criteria are constraining factors that represent the availability of infrastructure (road network, electricity connections, etc.), the existing charging network, and protected areas. Whereas demand criteria are influencing factors that reflect the attractiveness of each location centered on three different dimensions: residential, commercial, and leisure. The selection was bounded by two constraints: data availability and granular resolution.

As a first step for data processing, the datasets were preprocessed and reproduced separately as GIS layers. Then, a base grid layer is created where the study area is transformed into a hexagon-grid. This transformation enables location evaluation for the potential of EV charging in each cell (hexagon) within the total study area. The size of the hexagon needs to be determined on the account of providing a meaningful and conclusive location evaluation while maintaining its applicability for the different dataset resolutions. After that, the data values of each criterion were projected to the corresponding cells either by their sum, count, or average, depending on the layer type.

For supply criteria, public space availability and proximity to electric grid connections are critical considerations for locating charging stations. In addition, several preferences highlighted by the municipality—such as clearance from trees and closeness to streetlight posts—were considered. Thus, a spatial analysis is carried out for these GIS layers by buffering, intersecting, and excluding the supply criteria layers using the Euclidean distance method. The resulting layers are first joined and then projected to a base grid layer that mirrors the infrastructural availability in each cell. Wherever there is an availability, the cell is given a value of one, whereas its absence is zero.

As for demand criteria, layers were transformed into grid-layers by projecting their data to the corresponding overlaying cells either by sum or average, depending on each criterion type and resolution. In this way, a uniform projection system was established for

all layers, and their data values were captured on a cell level. To normalize data values and mirror the attractiveness of each cell in a given layer, a five-point scoring system was established for each. A frequency distribution was constructed from each dataset to determine the intervals of the five mutually exclusive classes. While zero corresponds to unattractive cells, five corresponds to the most attractive. The positive and negative correlation was reasoned by the authors and literature.

The next step was to obtain the suitability index map from the weighted sum of all the demand criteria layers subject to the supply constraints. Accordingly, a final base layer is produced, including the scorings of all the demand and supply criteria for each cell in one attribute table. To determine the weights of each demand criterion, the analytic hierarchy process (AHP) was applied. This mathematical method is one of the most popular multicriteria techniques for prioritizing and selecting alternatives in complex decision problems. It was originally proposed by Saaty [25] and has been widely employed in various real-world applications. Under the assumption of rational behavior, the AHP models a decision-making problem as a hierarchy of criteria, sub criteria, and alternatives. Being the municipality in this study, the decision maker gives the relative preferences over the set of selected demand criteria. The ability of transforming qualitative comparisons into numerical values is the main characteristic contribution of this method [26]. Through pairwise comparison matrices and using discrete relative measurements that are expressed in numbers ranging from 1 to 9, the weight of each criterion is calculated. Finally, the decision maker's opinion is tested for consistency based on Saaty's recommended consistency ratio of 0.10 or less.

The selection of this method was based on its subjectivity and ability to reflect the decision maker's preferences. Since the development of the charging infrastructure depends on the municipality's strategy, specifically if they intend to prioritize public charging for one demand dimension over the other (e.g., residents), this method serves the needs.

*3.3. Cluster Analysis*

In the final step, a cluster analysis was performed on the suitability index map to optimally allocate the derived charging stations across top-scoring clusters and ensure wider spatial coverage. This step stems from two objectives that public authorities might have other than profitability. First, to ensure a convenient and reliable geographic coverage rather than concentrating the charging stations in profitable areas. Second, as an attempt to reduce the public charging access gap for underserved communities with more equitable distribution across the clustered areas within the municipality. As several studies concluded that expansions in public charging networks mainly take place in privileged communities [27,28]. With the second-hand EV market maturing, this could be a remaining barrier for less privileged communities even though private charging is the main metric for infrastructure equity since it delivers low-cost charging [29]. Consequently, the final product of this study is a layout of the proposed volume of public charging stations precisely located within the study area.

**4. Case Study**

Located in the northern Ruhr district, formerly the industrial heart of Germany, Bottrop shares a history of coal mining that goes about 155 years. As of 2018, the last hard coal mine in the city and Germany came to an end as the city undergoes a structural change. With a population of 117,000 inhabitants, the city emerged as a promising example of a highly industrialized city that has combined a forward-looking transition process and ambitious $CO_2$ targets. Bottrop was previously selected for Innovation City Ruhr pilot which aims for an energy-efficient urban redevelopment, and has achieved 38% $CO_2$ emission reduction in only 5 years [30]. In continuous efforts to support these objectives, Bottrop's municipality aims to develop a concept for promoting electric mobility in collaboration with the Institute for Climate Protection, Energy and Mobility (IKEM). The following case study was carried

out in cooperation with IKEM. In this section, the application of the methodology presented in Section 3, along with the assumptions and calculations, are illustrated.

### 4.1. Demand and Finance Model

Starting with the energy balance equation presented in the methodology section Equation (1a).

The parameters $N$ (number of electric vehicles) and $R_i$ (share of public charging) are difficult to obtain from the current data. Since there is no certainty in projecting the evolution of EV fleet, as well as the percentage of charging taking place at public charging stations, four scenarios were developed for the year 2030 considering the EV adoption rate and the share of public charging. The size of the EV fleet in 2030 was estimated from two projections, namely a realistic one (progressive scenario) and an optimistic one (thriving scenario). The first is based on the polynomial regression of historical data of electric passenger vehicle registration provided by the German Federal Motor Transport Authority between 2017 and 2021 in Bottrop [31]. While the second relies on the German federal government ambitious goal of 10 million electric vehicle by 2030. Out of it, it is estimated that 9 million electric vehicles will belong to the passenger segment. The yearly developments of the EV stock in Germany to reach this target was modeled by the German Energy Agency [32]. Since this projection is for Germany, the EV per capita ratio was derived for each year using its population in 2020 as a base year [33]. The yearly ratios were then applied to the population of Bottrop taken on the same base year [34]. As a result, Figure 4 exhibits the development of the EV stock in Bottrop until 2030.

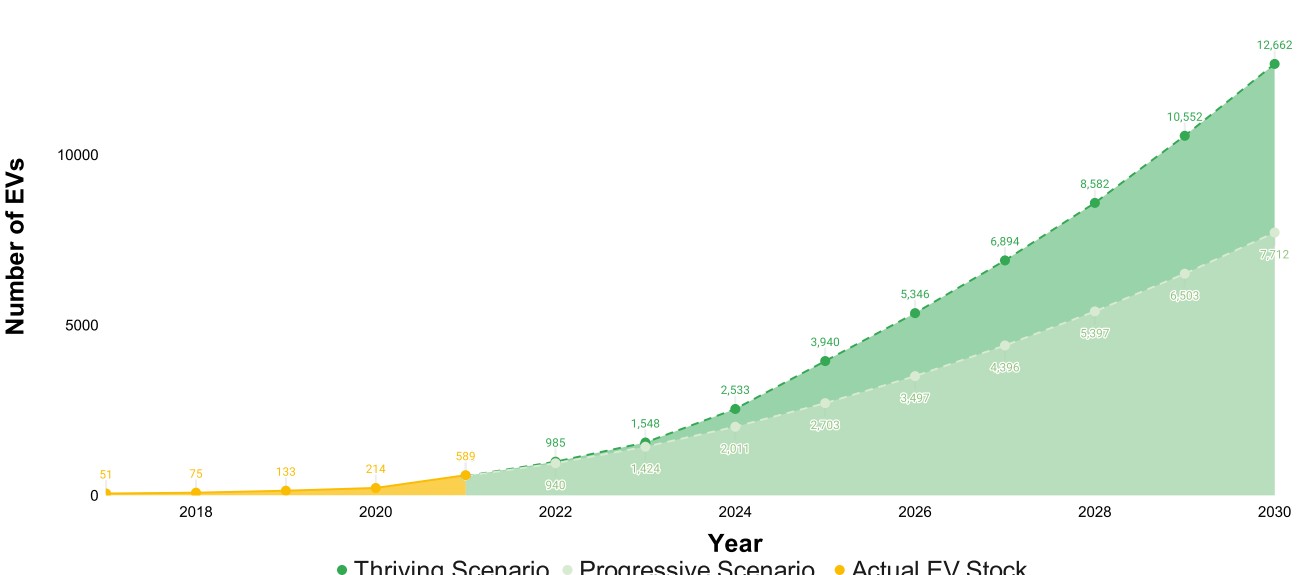

**Figure 4.** Projected passenger EV stock development in Bottrop.

For comparability, the total passenger vehicle stock in Bottrop was linearly projected from the vehicle registration history provided by the German Federal Motor Transport Authority between 2017 and 2021 to derive the EV adoption rate by 2030 [31]. Building on the forecast of the German federal government master plan for charging infrastructure, the ratios of non-public (home and work) to public charging will range from 60% to 40% and 85% to 15% respectively [35]. Consequently, the scenarios in Figure 5 are constructed:

| | | Scenario 1 | Scenario 2 | Scenario 3 | Scenario 4 |
|---|---|---|---|---|---|
| | | *Progressive & Private* | *Progressive & Public* | *Thriving & Private* | *Thriving & Public* |
| | EV adoption rate | 10% | | 16% | |
| | EVs - $N$ | 7,712 | 7,712 | 12,662 | 12,662 |
| | Share of public charging - $R_i$ | 15% | 40% | 15% | 40% |

**Figure 5.** Overview on 2030 scenarios.

The parameters $e, d,$ and $p_i$ can be deduced from the following data:

*e*: **energy consumption of an EV (kWh/km)**—The General German Automobile Club (ADAC), Europe's largest motoring association, conducts independent assessment to evaluate the environmental performance of cars holistically for prospective buyers. For electric vehicles and plug-in hybrids, ADAC Ecotest follows a special electric cycle consisting of an adapted WLTC along with a highway cycle under standardized conditions. Over 28 passenger electric vehicles of different classes were tested for energy consumption, energy required for full charge, and range. The results were also compared against the manufacturer specifications. For this parameter, the average energy consumption was calculated at 21.3 kWh/100 km equivalent to 0.213 kWh/km, while the average loss from recharging was 12%.

*d*: **daily driven distance (km)**—According to the Mobility in Germany (MiD) latest survey in 2017, the annual mileage estimated by the respondents averaged 14,700 km. This part of the survey covered exclusively passenger cars of private households and describes everyday mobility of the federal German resident population. This corresponds to a daily driven distance rounded to 40 km for a car owner. Although vehicles used in business fleets had nearly a double annual mileage compared to private passenger vehicles, it is not differentiated in this study. This stems from the fact that only a small part of 5 million commercial passenger vehicles were registered in 2017 compared to 41 million private passenger vehicles [36].

$p_i$: **power of a public CS**—The German Federal Ministry of Transport and Digital Infrastructure has launched funding programs to support the deployment of normal and fast charging infrastructure for private investors, cities, and municipalities. Although the formula allows multiple types of public chargers to be considered, only one type was selected in line with the municipality's preference. Hence, the selected public charging station in this study is assumed to have a 22-kW (AC) charging power.

As for the minimum utilization rate, the public investment case is considered which yields to Equation (7).

The values of the parameters: $c$, $t_a$, and $m$ are set as follows:

*c*: **capital investment (€)**—The total investment required for a charging station includes the product cost along with its installation, operational, and maintenance costs over its lifetime. According to Deloitte's business report on charging infrastructure in Germany, the average cost of a 22-kW public charging station—including the installation cost—is 7500€. As for operations and maintenance, the annual cost is 750€ [37]. Taking into account the current year-on-year average changes in producer prices of services [38], an adjusted operational and maintenance cost of 800 €/year was considered. The total investment was then calculated over a charging station's lifetime.

$t_a$: **period of amortization (years)**—Since the period of amortization is equal to the lifetime of the charging station, a lifetime period of 8 years was considered relying on the German National Platform for Electric Mobility study [3].

*m*: **marginal profit (€/kWh)**—For simplicity, the pay-per-use model was assumed. It only includes the price paid for the energy withdrawn from the charging station and excludes parking and additional service fees. Therefore, *m* is the difference between the e-mobility service provider price and the electricity cost. Covering different prices from

several providers in Germany in 2020, the average price is 0.319 €/kWh [39]. As reported in the German Federal Association of Energy and Water Management electricity price analysis in January 2021, the average electricity price for industry is 0.1825 €/kWh, including electricity tax [40]. As a result, *m* was calculated at 0.1365 €/kWh and assumed to remain constant throughout the study period.

To summarize, Table 1 lists the above-mentioned assumptions and parameters used as inputs for the energy balance equation.

**Table 1.** Key parameters and assumptions by scenario.

| | | Scenario 1 | Scenario 2 | Scenario 3 | Scenario 4 |
|---|---|---|---|---|---|
| | | *Progressive and Private* | *Progressive and Public* | *Thriving and Private* | *Thriving and Public* |
| **Inputs** | **Unit** | **Values** | | | |
| $N$— number of electric vehicles in 2030 | count | 7712 | 7712 | 12,662 | 12,662 |
| $R_i$— share of public charging | % | 15 | 40 | 15 | 40 |
| $e$— energy consumption of an electric vehicle | kWh/km | 0.213 | 0.213 | 0.213 | 0.213 |
| $d$— daily driven distance | km | 40 | 40 | 40 | 40 |
| $p_i$— charging power of the selected CS | kW | 22 | 22 | 22 | 22 |
| $U_i$— utilization rate | % | 7 | 7 | 7 | 7 |
| $c$—captial investment | € | 13,900 | 13,900 | 13,900 | 13,900 |
| CAPEX | € | 7500 | 7500 | 7500 | 7500 |
| OPEX | €/year | 800 | 800 | 800 | 800 |
| $t_a$—period of ammortization | year | 8 | 8 | 8 | 8 |
| $m$—marginal profit | €/kWh | 0.1365 | 0.1365 | 0.1365 | 0.1365 |

*4.2. GIS Model*

4.2.1. Criteria Selection and Data Collection

After conducting the literature review and based on experts' input, the evaluation criteria for optimal locations of EV charging stations were identified and selected. The selection was bounded by two constraints: data availability and granular resolution. As a result, 16 criteria were identified in the case study and classified into supply and demand criteria. The supply criteria represent the availability for charging infrastructure and the spatial constraints in the study area. For this study, there are three criteria related to infrastructure availability. Regarding the spatial constraints, another three main criteria were analyzed. The demand criteria reflect the attractiveness of the location for a charging station and the likelihood for potential charging demand driven by higher EV adoption possibility. The demand criteria are clustered in three main dimensions (residential, commercial, and leisure). Table 2 lists the criteria, their resolution and source.

4.2.2. Data Processing

First, the datasets of the 16 criteria were examined in the form of GIS layers. Data cleaning was then performed to remove incomplete, duplicate, or irrelevant data from supply and demand dimensions. To protect confidential information, only the processed data are presented. Next, a base layer is created where the study area is divided into a 100 × 100 m hexagon-grid (vertical and horizontal spacing). This transformation was conducted to enable the evaluation of the potential of EV charging infrastructure in each cell within the total area of Bottrop as illustrated in Figure 6. The size of the hexagon cell was determined on the account of providing a meaningful and conclusive location evaluation while maintaining its applicability for the different dataset resolutions. After that, the data values of each criterion were projected to the corresponding cells either by their sum or average, depending on the layer type (which are described in the section below).

**Table 2.** List of criteria by type of analysis.

| | Dimension | Criteria | Analysis | Resolution | Source |
|---|---|---|---|---|---|
| Demand | Residential | Households | Count | Unit | Municipality |
| | | Type of buildings | Average | Building block | Municipality |
| | | Buildings | Count | Unit | OSM |
| | | New residential developments | Count | Unit | Municipality |
| | | Long-term unemployment | Average | District | Social atlas |
| | | Vechicle registrations | Count | District | Municipality |
| | Commercial | Companies | Count | Unit | OSM |
| | | Traffic | Average | Street | Municipality |
| | | Parking lots | Count | Unit | OSM |
| | Leisure | POI | Count | Unit | OSM |
| Supply | | Road network | Euclidean distance | Street | OSM |
| | | Electricity grid connections | Euclidean distance | Unit | Municipality |
| | | Existing charging stations | Euclidean distance | Unit | Municipality |
| | | Streetlights | Euclidean distance | Unit | Municipality |
| | | Protected Areas | Euclidean distance | Unit | Municipality |
| | | Trees | Euclidean distance | Unit | Municipality |

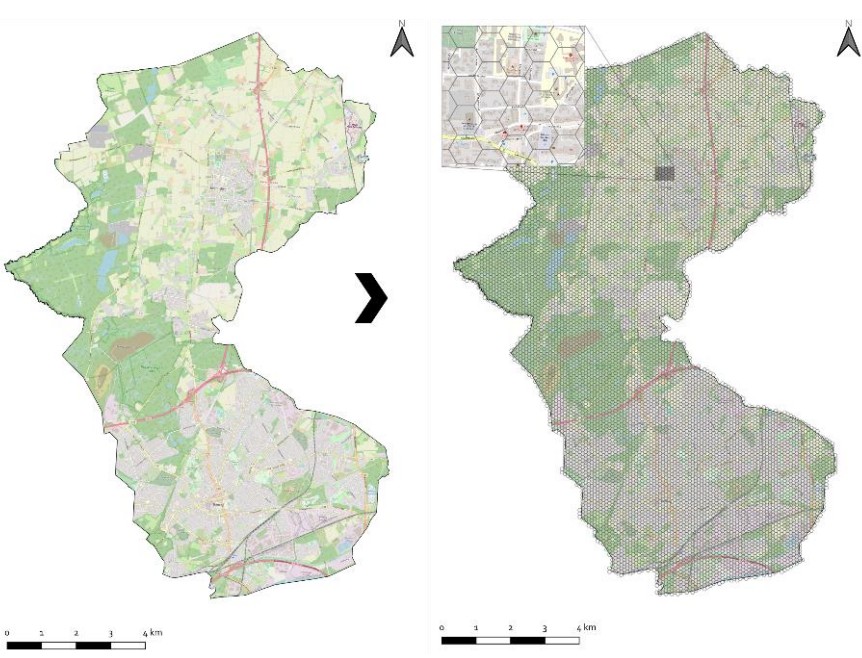

**Figure 6.** Creation of the base grid layer of the study area (Bottrop).

Supply Criteria

The geo-analysis for supply criteria is conducted by either the intersection of these criteria, or their exclusion.

Starting from the three criteria related to infrastructure availability, a buffering and intersection analysis was applied. First, the road network comes naturally as a fundamental requirement for electric vehicles to access and to park on the sideways for recharging. Since the dataset is formed of polylines, a 50 m buffer width was applied to ensure a full coverage of the road and the sideways. Second are the existing electricity grid connection points provided by the municipality to ensure that the locations of the charging stations do not require a major expansion of the grid network which entails excessive investments. A conservative and reasonable buffer radius of 15 m was set to minimize cable extension work and cost compared to 50 m from a study by Gkatzoflias et al. [17]. The capacity and type of the existing connection points were not provided for this study and might require a separate case-by-case evaluation from the distribution system operator (DSO) if it needs

to be upgraded. Such data can have a critical impact on location selection where high investments on the grid are needed. Therefore, this falls outside the scope of this study. As a preferable recommendation by the municipality, the third criterion is the proximity from streetlight poles. The rationale behind it was to offer more visibility for the driver to spot the charging station while driving and later to use it. Accordingly, the location points provided were buffered with a radius of 10 m to resemble the lighting coverage of a streetlight post. The final spatial analysis step for the supply criteria, ensures that they coincide by performing a geometric intersection across the three layers. The resulting layer is a representation of all the areas where infrastructure is available.

Regarding the spatial constraints, protected areas such as historic sites and environmental zones were excluded from the infrastructure availability layer produced above. The second constraint is related to the preservation of trees located in public areas. The point layer provided includes the possible crown radii of trees which were used as buffer radii for exclusion, assuming they are equal to the root crowns to avoid damaging the roots when constructing a charging station. The third constraint is the existing charging infrastructure which might lead to oversupplied service areas as the city currently hosts 31 public and semi-public charging stations. In consultation with IKEM, it was assumed that a normal-charging station has a service area that lies within a 300 m radius, whereas a fast-charging station's service radius was assumed at 400 m as an educated guess. Accordingly, buffer zones were created and excluded. Finally, the remaining areas represents the final areas of infrastructure availability and were projected to the grid base layer. Wherever those areas exist in a cell, a value of one is attributed to it, otherwise a zero values is applied. Figure 7 shows the finalized layer for infrastructure availability.

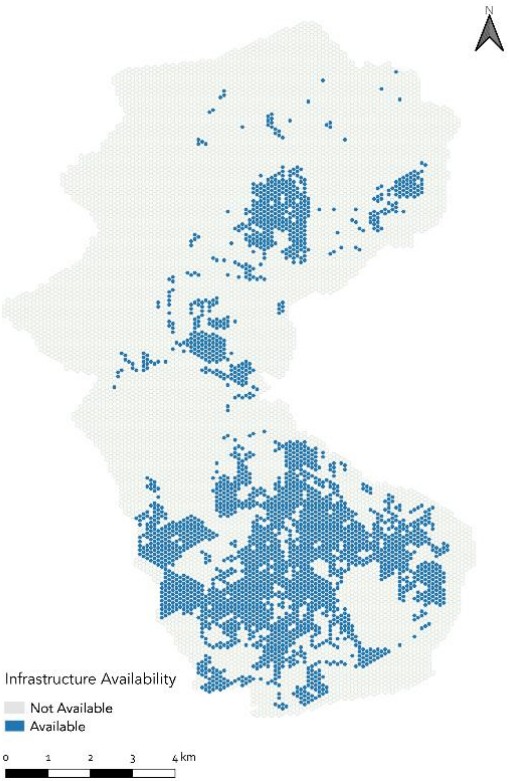

**Figure 7.** Infrastructure availability due to the fulfillment of all supply criteria.

Demand Criteria

The geo-analysis for demand criteria is conducted by sum or average depending on each criterion type and resolution. Layers were transformed into grid-layers by projecting their data to the corresponding overlaying cells. The following represents a description of each criterion and how it was projected to a separate base grid layer:

*Residential Criteria:*

**Households:** The number of households offers an improved overview of the population density since it is not limited by age groups. The attractiveness and potential of a charging station's location is greater where the household density is higher. In this study, the municipality provided the most recent household distribution on a statistical district level.

**Type of buildings:** The type of building can indicate the likelihood of home charging or public charging. Houses of single-family detached type often have a parking garage or space, hence a higher prospect to install private chargers. On the other end, apartment buildings tend to rely more on roadside and public parking lots, hence a higher prospect for public charging. The available data provided by the municipality from the land registry data provide the average number of floors on building block level. Although a better representation could be obtained if municipalities could collect data on housing types, a correlation between number of floors and type of buildings was adopted. Hence, the evaluation was based on the number of floors instead. The higher the average number of floors is in a given building block, the more attractive it is for the location of a public charging station.

**Buildings:** The number of residential buildings was considered an indication of the level of utilization and the local coverage of a charging station. The more buildings there are, the more the charging station be utilized, and the better it can serve the charging demand. The residential building data were extracted on a single-building level from OpenStreetMap which enabled a higher resolution analysis.

**New residential developments:** In line with the development plan of charging infrastructure in 2030, the new residential developments in the city were considered. The current development projects for residential units are represented by the number of buildings and units in the construction phase at a building resolution. It relies on the data provided by the municipality from the residential real estate developers. The number of units was selected to evaluate the volume of new developments which is positively correlated to the potential locations of charging stations.

**Long-term unemployment:** A survey conducted on EV adopters in Germany revealed that EV users have significantly higher incomes than drivers of conventional cars, adding that 70% of respondents were in full-time employment [41]. It was concluded that the economic condition has a positive and strong correlation to car ownership and in particular EV adoption. Consistently, the Mobility in Germany (MiD) survey in 2017 stated that lower incomes and lower employment rates play a role in car ownership in Germany. In addition, it was found that nearly half of the mileage of passenger cars is made in the course of commuting to work or business activities [36]. It is worth noting that this criterion counters the equitable deployment of public chargers and widens the access gap in favor of privileged communities. Its implications should be carefully understood by decision-makers while rating it. This criterion, provided by the municipality, has a close and negative correlation to EV adoption and, consequently, to charging stations' potential locations.

**Vehicle registrations:** The number of cars registered in each statistical district provided by the municipality helps determine the city's passenger vehicles distribution. This can lead to a better allocation of charging stations by factoring in multiple passenger car ownership of households and companies. In addition, since private vehicles in most cases are underutilized assets, this criterion marks the areas where the corresponding vehicles will be parked for most of their time. Therefore, a positive correlation can be drawn from the number of vehicles registered and the potential location of charging stations.

*Commercial Criteria:*

**Companies:** The number of companies can provide a valuable indication for two main use cases. The first use case is for the company's customers, while the second is for their employees who might use nearby street or public parking for opportunity charging.

**Traffic:** Besides characterizing traffic flows and volume for maximizing road network efficiency, traffic analysis offers a valuable insight on the attractiveness of the charging station location model. Roads with high traffic volume inherently generate higher demand on charging. In addition, charging stations located on these roads have increased visibility, which in turn drives more positive EV perception. The hourly peak volume of motor vehicle traffic in relevant roads was analyzed and classified according to an ordinance called RASt-06 (Richtlinie für die Anlage von Stadtstraßen) published by the Research Association for Roads and Traffic in Cologne [42]. The classification resulted in five levels of traffic flows ranging from low to high.

**Parking lots:** Public charging takes place mainly in street parking or parking lots. While the first is accounted for in the supply criteria as a fundamental requirement, the latter emerges as an attractive factor for locating a charging station. Since the development of charging infrastructure is on a municipal level, only public parking lots were considered, excluding the possibility of semi-public charging infrastructure.

*Leisure Criteria:*

**POI**: Unlike traditional gas refueling, EV users will recharge their vehicles in areas where they spend time while their vehicles are parked. Hence, points of interest are naturally a prominent part of the planning of public charging network. Activities related to private errands, recreation, or shopping are plausible opportunities for EV users to recharge their vehicles publicly [9]. Table 3 shows the compiled list of POIs that were considered in this analysis, which were extracted from OpenStreetsMap data.

**Table 3.** List of POI.

| POI | |
| --- | --- |
| Malls and department stores | Theme parks |
| Supermarkets | Parks |
| Museums and art centers | Sports and fitness centers |
| Theatres and cinemas | Hospitals and clinics |
| Conference centers and events venues | Educational institutions |

After projecting all the demand criteria and generating the corresponding base grid layers, a scoring scheme was applied to normalize the different dataset ranges. This step aims to bring all criteria into proportion by transforming their values into a specific range. To reflect the suitability of each cell, all demand criteria values were transformed to a five-point scoring system. Five mutually exclusive classes were constructed specifically for each criterion based on its frequency distribution, transforming all ranges from zero to five. An exception was made for parking lots and POIs' base grid layers since only their availability was considered in the analysis due to the absence of valuable data to rate them. Therefore, two mutually exclusive classes were constructed on the same range. While the availability of the criterion is rated at '5', the absence of it is '0'. After this step, every base grid layer has the same score range from '0' to '5' in each cell, positively correlated to the attractiveness of the location for a public charging station. A detailed overview on the scoring of each criterion is provided in the Supplementary Material.

*4.3. Analytical Hierarchy Process (AHP) Method*

To reflect the attractiveness of each cell, a weighted sum of all the demand criteria is calculated. This factors in the importance of one criterion over the other and provides a merited representation of the suitability scores. In this study, the AHP hierarchy structure shown in Figure 8 is comprised of the three demand dimensions mentioned earlier. This level enables the municipality to prioritize their charging infrastructure provision between these dimensions. The 10 demand criteria in Table 2 are clustered under each dimension. This level compares the relative importance of each criterion to that of the same dimension.

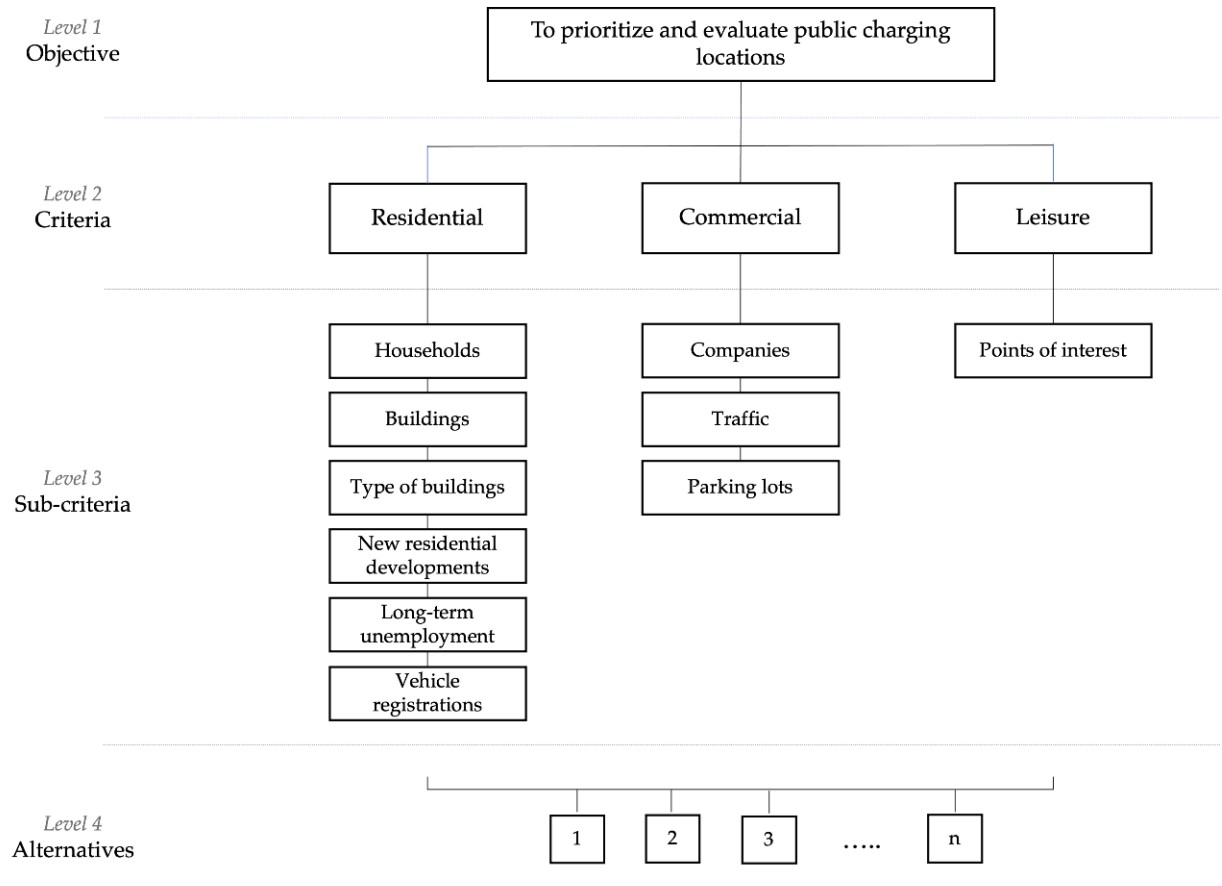

**Figure 8.** AHP hierarchic structure of the demand criteria.

To obtain the weights, a questionnaire was given to the municipality's mobility manager to evaluate the designed pairwise comparison tables by using the method's standard scale. A detailed overview on the questionnaire is presented in the Supplementary Materials. It is noted that the inconsistency ratios, which indicate the consistency of the evaluation, are higher than the ideal ratio of 0.1. The pairwise comparison matrices for the dimensions and criteria are given in Tables 4–6 along with the calculated weights.

Finally, the overall weights of the criteria combined with the dimension's weights are summarized in Table 7 and Figure 9 below.

**Table 4.** Weights and pairwise comparison matrix of demand dimensions.

| Dimension | Residential | Commercial | Leisure | Weight |
|---|---|---|---|---|
| Residential | 1 | 9 | 9 | 79.7% |
| Commercial | 1/9 | 1 | 5 | 15.1% |
| Leisure | 1/9 | 1/5 | 1 | 5.2% |
| | | CR = 0.308 | | |

**Table 5.** Weights and pairwise comparison matrix of commercial criteria.

| Commercial | Companies | Traffic | Parking Lots | Weight |
|---|---|---|---|---|
| Companies | 1 | 6 | 1/7 | 20.9% |
| Traffic | 1/6 | 1 | 1/5 | 7.1% |
| Parking lots | 7 | 5 | 1 | 72.0% |
| | | CR = 0.549 | | |

**Table 6.** Weights and pairwise comparison matrix of residential criteria.

| Residential | Households | Buildings | Type of Buildings | New Residential Developments | Long-Term Unemployment | Vehicle Registrations | Weight |
|---|---|---|---|---|---|---|---|
| Households | 1 | 5 | 5 | 1 | 9 | 1/2 | 31.1% |
| Buildings | 1/5 | 1 | 1/2 | 1 | 9 | 1/2 | 10.7% |
| Type of buildings | 1/5 | 2 | 1 | 1 | 9 | 1 | 15.2% |
| New residential developments | 1 | 1 | 1 | 1 | 9 | 1 | 17.4% |
| Long-term unemployment | 1/9 | 1/9 | 1/9 | 1/9 | 1 | 1/9 | 1.9% |
| Vehicle registrations | 2 | 2 | 1 | 1 | 9 | 1 | 23.7% |
| CR = 0.104 | | | | | | | |

**Table 7.** Overall weights of criteria.

| Criteria | Weight |
|---|---|
| Households | 24.79% |
| Vehicle registrations | 18.89% |
| New residential development | 13.87% |
| Type of buildings | 12.11% |
| Buildings | 8.53% |
| Long-term unemployment | 1.51% |
| Parking lots | 10.87% |
| Companies | 3.16% |
| Traffic | 1.07% |
| POI | 5.20% |

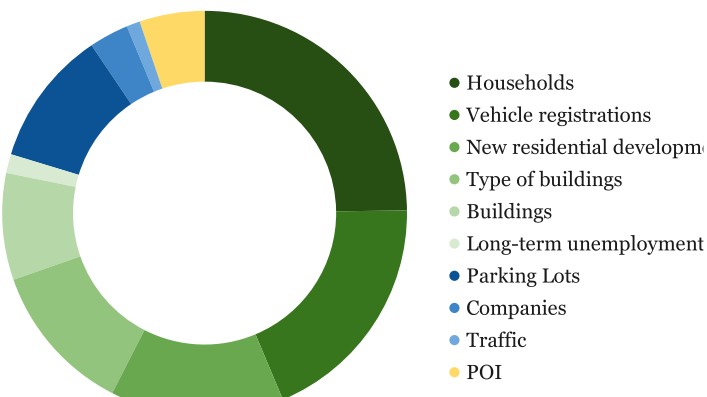

- Households
- Vehicle registrations
- New residential developm
- Type of buildings
- Buildings
- Long-term unemployment
- Parking Lots
- Companies
- Traffic
- POI

**Figure 9.** Overall weights of criteria.

It is evident that the municipality strategy is focused mostly on the residential dimension, whereas the commercial dimension comes as a second priority. The leisure dimension had the least priority compared to the other dimensions. To obtain the suitability index map, the scores of all the demand criteria grid layers are summed up according to these weights and multiplied by the binary values of the infrastructure availability layer in Figure 7. To this end, the suitability index map mirrors the attractiveness of all locations in the city through the same grade scale from '0' to '5'. The usefulness of this map is that it not only enables the strategic expansion of the charging network developed in this study, but it also serves as a key evaluation for demand-driven expansion.

## 5. Cluster Analysis

In the final step of this study, the suitability index map is clustered using k-means clustering algorithm. This algorithm is popular for automatically grouping points or polygons into a predefined number of clusters based on their 2D distances from each

other. It runs first with a group of randomly assigned centroids equal to the selected number of clusters and is used as initial points for every cluster. Then it iterates on the calculations of the 2D distances between the data points and the centroids to optimize their positions. Each iteration re-adjusts the centroids' location until the centroids stabilize and the clustering is successful, or when the defined number of iterations is reached. To ensure a wide geographical coverage, the number of clusters was set to 30 as shown in Figure 10. Each color stands for a unique cluster. The algorithm takes the centroid of each $100 \times 100$ m hexagon cell, and groups all of the cells from the suitability index map into 30 clusters. After setting the number of clusters to 30, the k-means algorithm converged after 24 iterations and was successful.

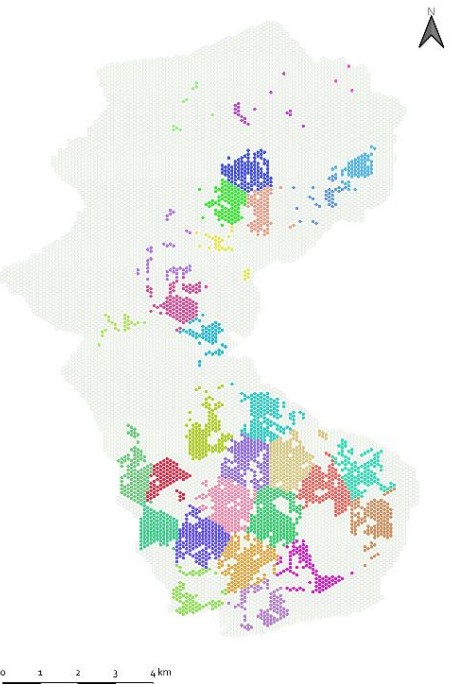

**Figure 10.** K-means clustering algorithm output (N = 30).

Next, the number of charging stations derived in each scenario is distributed proportionally on the clusters. Based on the average score of the cells within each cluster over the sum of all average cluster grades, the share is calculated and the number of charging station within each cluster is derived. After allocating the number of charging stations per cluster, the charging stations are assigned to the top scoring cells within each. The result is a layout of the proposed number of charging stations located specifically on the map for each scenario.

## 6. Results

The following section showcases the results of the three primary steps presented in this study. First is the estimation step which derives the number of public charging stations needed by 2030 and is established on the four constructed scenarios. Second is the location step, representing the GIS analysis of the residential, commercial, and leisure dimensions followed by the suitability index map. Subsequently, the optimization step—rendering the cluster analysis—outlines each scenario's proposed charging infrastructure network.

### 6.1. Estimation

Table 8 exhibits the output of the demand and finance model for the four constructed scenarios. For comparability, the ratio of EV per public charging station is exhibited in this table. In the early development stage, the demand for charging is a complex and unpredictable evolution. As a result, two common approaches that fall under the same

push strategy, come into play: political targets or empirical studies. Political targets, often initiated by political actors, rely on setting a ratio of EVs per public charging point to achieve envisioned future projections of EV uptake and related policy goals.

**Table 8.** Estimated number of public charging stations per scenario.

|  | Scenario 1 | Scenario 2 | Scenario 3 | Scenario 4 |
|---|---|---|---|---|
|  | *Progressive and Private* | *Progressive and Public* | *Thriving and Private* | *Thriving and Public* |
| EV adoption rate $N$ | 10% | | 16% | |
| Share of public charging $R_i$ | 15% | 40% | 15% | 40% |
| Proposed number of public CS | 323 | 862 | 517 | 1379 |
| Ratio of EV per public CS | 24 | 9 | 24 | 9 |

For instance, the Alternative Fuels Infrastructure Directive recommends for EU member states an ideal ratio of a maximum of 10 PEVs per charging point [43]. In China, the National Development and Reform Commission (NDRC) recommends a ratio of 8 to 15 PEVs per charging point; whereas in the U.S., different ratios of 7–14, 24, and 27 are benchmarked by different sources. Nevertheless, the effectiveness of these ratios to meet the future charging demand remains uncertain [44]. Such policies primarily aim to ensure a minimum coverage level to encourage a greater range of confidence and to promote PEV awareness, even under low utilization rates. Remarkably, the calculated set of ratios in this study falls in line with the mentioned targets.

While Scenario 1 represents the least aggressive uptake on both EV adoption rate and the share of public charging, Scenario 4 is the most aggressive. This is reflected by the calculated number of charging stations. For a lower share of public charging, the recommended number of public charging stations is 323 in Scenario 1 and 517 in Scenario 3, while the EV adoption rate increases from 10% to 16%. The recommended number of public charging stations increases to 862 in Scenario 2 and 1379 in Scenario 4 with the higher share of public charging, while the EV adoption rate increases from 10% to 16%. Comparing Scenarios 1 and 2, the higher share of public charging from 15% to 40%, increased the number of charging stations by a factor of around 2.7. Similarly, Scenarios 3 and 4 maintain the same factor. Given the narrow corridors in EV adoption rate between the scenarios, the rise from 10% to 16% between Scenario 1 and 3 increased the number of charging stations by a factor around 1.6. The same applies between Scenarios 2 and 4 as well.

Moreover, the finance model indicates a utilization factor of 7% across all scenarios since the same public charging station characteristic, cost, and revenue are assumed. The utilization factor calculated is based on break-even over the lifetime of the charging station and excluding the cost of capital. This yielded a low utilization factor and had amplified the number of charging stations derived. For a higher utilization factor, driven by introducing profits and costs, the number of charging stations will eventually decrease.

*6.2. Location*

Being determined only on demand and supply criteria, the GIS analysis is independent of the scenarios developed earlier. Hence, it is applicable for all the scenarios. Figure 11 illustrates the scores of each dimension after applying the weighted sum of scores of the corresponding criteria.

The residential dimension scores show a wider distribution across rural and suburban areas of the city with higher values concentrated particularly in the northern part due to its residential nature. The commercial dimension scores are dispersed across the city and concentrated in the southern part as it revolves around the city's commercial center and suburban areas. A similar pattern can also be noticed for the leisure dimension. Since this dimension has only one sub-criterion, the weighted score is equivalent to the total sub-criterion score. As mentioned in the case study section, the scoring of the points of interest criterion is based only on the availability of POIs which is mirrored in its scoring range.

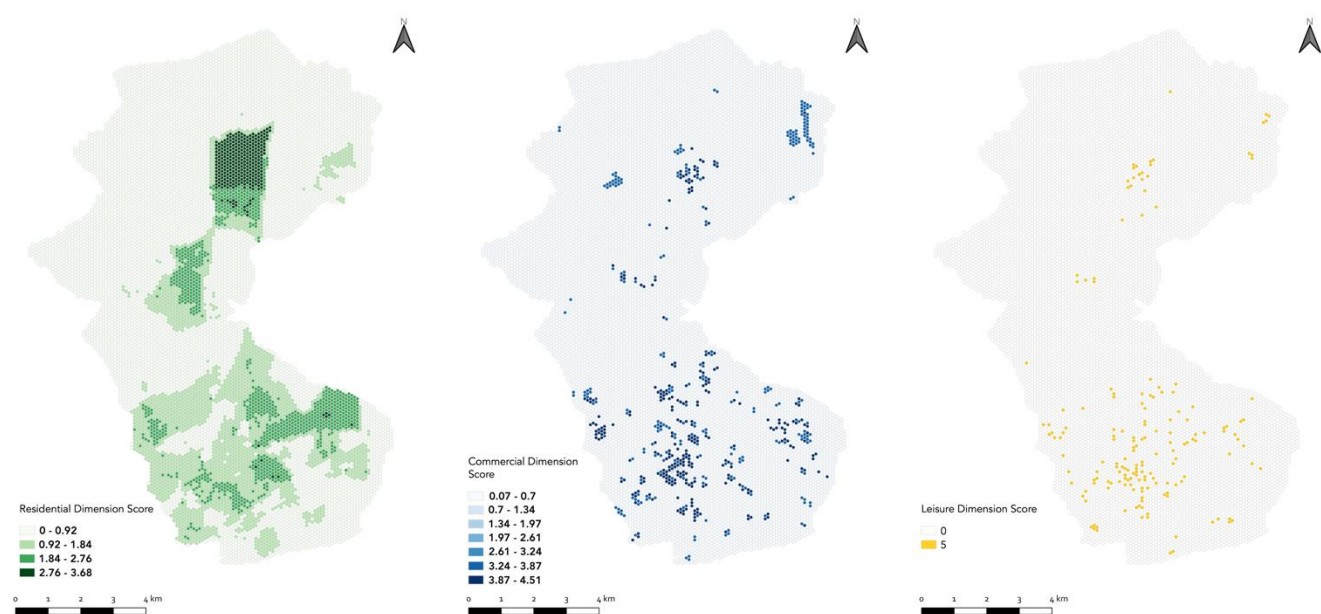

**Figure 11.** Weighted scores of demand dimensions.

Considering the weights of each dimension, the weighted sum of scores of the three dimensions yields to the final total scores. Combined with the supply constraints depicted by the infrastructural supply layer, the suitability index map is produced as shown in Figure 12. As deduced from the mobility manager's input, the highest preference was given to the residential dimension, leading to higher scores particularly in areas that overlap with the residential dimension map; notably, in the northern part of the city. This highlights the significance of the cluster analysis conducted in the next step to ensure a wide geographical distribution whilst giving precedence to the cluster's top-scoring cells.

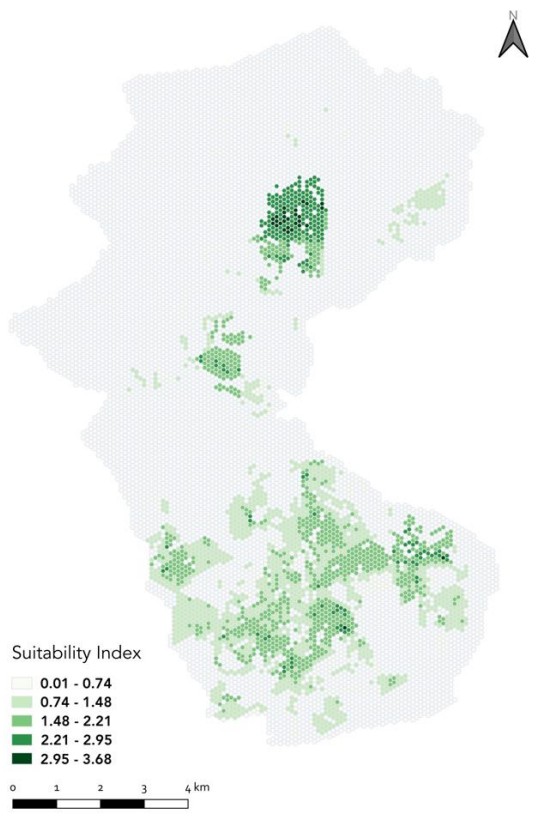

**Figure 12.** Suitability index map.

The suitability index map is an overall reflection of the location attractiveness for public charging stations covering all the city areas. It provides a reference for the municipality to build and expand its charging network in an effective and adaptable way. On the other hand, the municipality could also benefit from it to evaluate demand-driven public charging networks. It can serve as a primary evaluation tool for the demand requested and assist in prioritizing the most suitable locations when resources are limited.

### 6.3. Optimization

After running the k-means clustering algorithm, the suitability index map was classified into 30 clusters. The average scoring of each cluster was calculated and weighed against the total sum of average scores of all clusters. Each cluster was then assigned a share of the total proposed public charging stations in each scenario (see Table 8). Once the share for each cluster is obtained, the charging stations were assigned on a one-to-one basis to the top-ranking cells of each cluster. Only in Scenario 4, which is marked by the highest number of charging stations, three clusters were saturated as their share of allocated charging stations exceeded the number of cells they contain. In this case, a second iteration was made on the top-ranking cells where a second charging station was assigned in addition to the first one until the total share is completely attained. Figures 13–16 show the final layout for each scenario after the optimization step. The charging stations are located at the center of every cell. Knowing that each cell fulfills all the constraints set in the supply dimension, a site visit and an assessment of the electric connection points capacity are needed to precisely locate and build the charging stations in their corresponding cells.

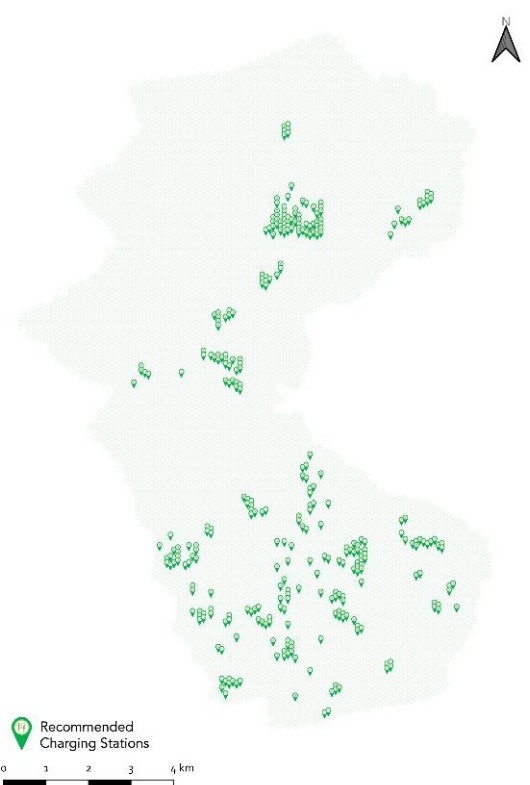

**Figure 13.** Scenario 1: Public charging network layout.

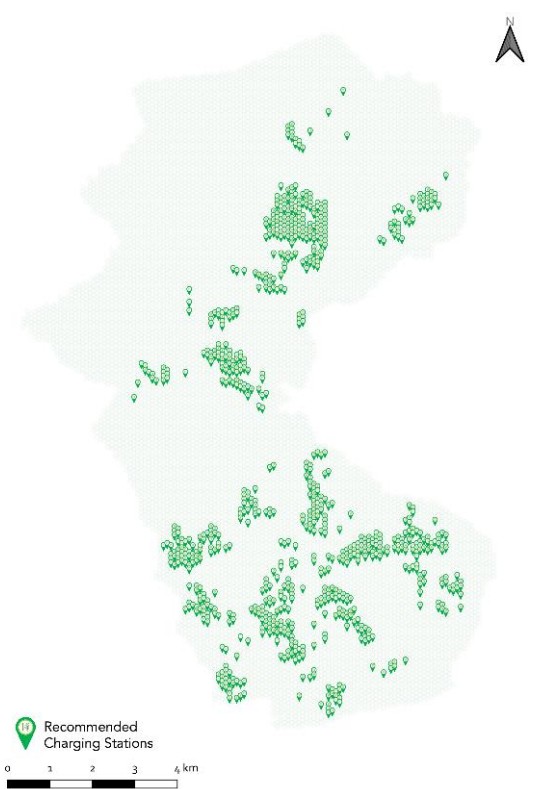

**Figure 14.** Scenario 2: Public charging network layout.

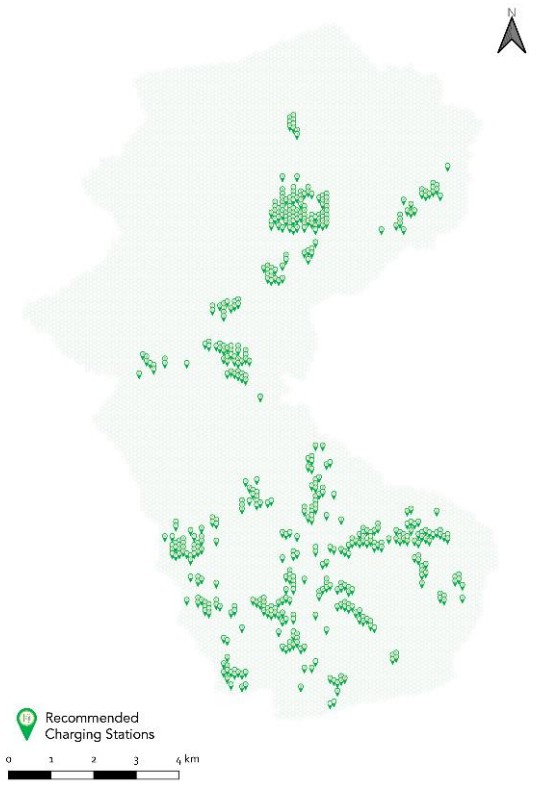

**Figure 15.** Scenario 3: Public charging network layout.

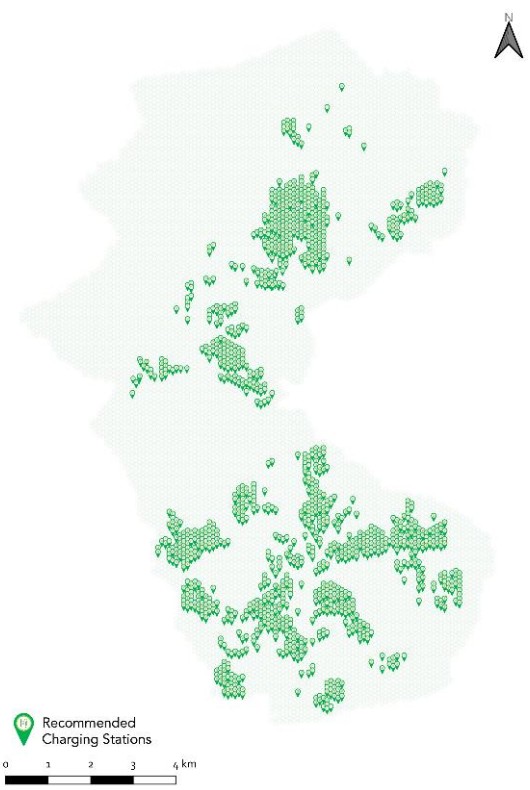

**Figure 16.** Scenario 4: Public charging network layout.

For a responsive and feasible strategy, it is recommended that the municipality plan its charging network according to scenario 1 and monitor the EV adoption rate development and the degree of utilization of the installed charging stations. If the uptake follows a rapid growth trend, then scenarios 2, 3, and 4 can be adopted as a response measure.

## 7. Discussion

In line with the research questions presented, this study addressed the estimation of the demand for public charging stations on a municipal level and proposed a location model to plan the charging network. The methodology enables straightforward customization of input parameters, making it flexible and replicable for other cities. Nevertheless, it is also built on the strategy that the municipality intends to implement in order to build its charging network.

Looking ahead to 2030, the EV landscape is set for a progressive development induced by several dynamics. Xue et al. emphasized in their study of 20 countries that income, which mirrors the affordability of owning an EV, is one of the strongest drivers for EV market share [45]. With the continuous advancements in EV design and the economies of scale in production, the cost gap is expected to narrow as companies spread their costs more effectively, making EVs more attractive to consumers. Changes in energy prices and the ever-growing share of renewables in electricity generation are also giving electricity an edge over conventional fuel. In addition, government incentives—such as purchase subsidies and tax benefits—have proven to have a direct positive impact on EV uptake in the short run [46]. While such dynamics draw a positive trend for EV adoption, it is still ambiguous how aggressive the trend will be. Another major hurdle lies in the charging behavior of EV users in 2030. While home and work charging offer more convenience when available, public charging is challenged to compete. Moreover, the Building Electromobility Infrastructure Act (GEIG)—which came into force on the 25th of March 2021—stipulates that newly constructed and renovated residential and non-residential buildings must be equipped with connection infrastructure and recharging points with a larger number of parking spaces which will eventually encourage private charging. On the contrary, the

swift, wide, and reliable provision of public chargers might shift the consumer perception away from owning a private charger, leading to a higher share of public charging. In an attempt to broadly address these uncertainties, different EV adoption rates and shares of public charging were outlined in the scenarios presented in this study.

Another level of detail can be further investigated in the parameters of the energy balance equation. For instance, the driving behavior of EV users is assumed to be similar to that of conventional passenger vehicle users. In a study that surveyed early EV adopters in Germany, it was found that the average annual mileage varied between users of BEVs (10,300 km), PHEVs (13,600 km), and conventional (14,700 km) [41]. However, it was also highlighted that some users tend to use another car to carry out their trips. Since the aim should be to replace conventional vehicles with EVs, the conventional mileage was considered in this study. A potential refinement could be factoring in the PHEV pure electric mileage which requires modeling the PHEV driving profile and depends on its uptake versus BEVs. It is worth mentioning that the COVID-19 pandemic had profoundly impacted individual mobility behavior in Germany [47]. It accelerated many trends, such as remote working/studying and online shopping, which substantially reduced people's mobility. However, it is still unknown if this short-term change will translate into a permanent one.

Regarding the average energy consumption of EVs, only the existing models in the German market by ADAC were considered in this study. The vehicles varied in their classes and consequently in their energy consumption. For simplification, the average energy consumption did not factor in the dominance of one class over the other and took the overall average. Therefore, room for improvement exists in taking a weighted average of the forecasted shares of passenger car classes, as well as the anticipated advances of the electric powertrain technology in terms of energy efficiency. On the other side, the selection of the charging station power is determinant for deriving the demand and stretches to affecting the utilization factor. Notably, the energy balance equation allows the use of different power-level charging stations when the shares are given.

The utilization factor was derived from the finance model which covers the cost structure and the expected revenue stream of a public charging station. As mentioned earlier, the utilization rate derived was 7% which could be considered a minimum threshold to at least break even. It is primarily dependent upon the type of charging station selected, and its characteristics. The changes in cost structures and revenue streams also has a major influence on the utilization factor and thus on the derived number of charging stations. For example, a profit-driven investment will naturally require a higher utilization factor in order to compensate for additional costs associated with it, as well as to generate the expected return on investment. In this case, the costs of capital, business operations, and marketing should be factored in. Additional revenue streams—such as session fees, parking charges, and advertising—could also be considered. Furthermore, different business models—such as a subscription model—can be implemented in the calculations to fine-tune the utilization factor. Such investment is not limited to private companies, as municipalities could benefit from making profits to fund local EV incentive programs.

The presented geospatial analysis approach offers the scalability and flexibility to add as many criteria layers as desired. It provides a tailored solution that preserves the local characteristics of the city while focusing on the key determinants for potential public charging station locations. The availability of geo-data sets and their resolution was one of the limitations of this study. From the supply side, the availability of the electricity grid capacity and characteristics of the grid segments could have enabled the identification of capacity limits in each area and refined the overall infrastructural availability layer. From the demand side, few relevant socio-economic data were not available but could have a possible contribution. For example, age, income, and education levels were found to be relatively homogeneous for early EV adopters in Germany [41]. However, this could lead to a distorted analysis as the diffusion of EVs progresses to the critical mass which has heterogeneous characteristics. A higher and consistent resolution across all demand criteria layers could have contributed to more accurate analysis and results. The scoring system

permitted an explicit comparison between the dimensions and criteria layers through normalization and weighting. The input of the mobility manager had prime importance on the prioritization step of the suitability index map, which favored the residential dimension. As a matter of course, it is crucial to incorporate the inputs of key stakeholders as it immensely shapes the final results.

In the final step, the clustering algorithm played an integral role in ensuring the geographical distribution of the charging network. It prevented an excessive concentration of charging stations in the top-scoring cells which belongs mainly to the residential dimension. Instead, clustering enabled the localization of top-scoring cells in every cluster. Ultimately, selecting the rules for defining the clusters is subjective and can be altered depending on the level of coverage intended.

## 8. Conclusions and Outlook

The global transition towards electric mobility continues to gain momentum as electric vehicle sales are surging. Charging infrastructure and EV sales mutually influence each other and the need to match demand and supply through effective targeted investments plays a key role in reinforcing this cycle. To assist municipalities in planning their public charging network, a framework that combines demand and geospatial models was developed. As an exemplary case, a German city was the subject of this investigation. Four scenarios were introduced to address the possible development paths that EV adoption and charging behavior might follow in 2030. The respective paths were explored given the local preferences and assumptions from the industry. As a result, the estimated number of the required public charging stations varied from 323 in the base scenario to 1379 in the most aggressive scenario. This was based on a specific charger type, as well as a break-even finance model. Under each scenario condition, if more charging stations are to be installed, the charging network will be underutilized and therefore cannot be economically viable.

To identify the potential locations for public charging stations, a geospatial analysis was conducted to analyze the supply and demand factors within the municipality. Six supply criteria represented the infrastructural availability for charging stations and represented the study area's spatial constraints. On the other hand, 10 demand criteria which reflect location attractiveness, were categorized on three dimensions: residential, commercial, and leisure. A scoring system was developed to normalize the data and to allow for comparison. Based on the municipality's mobility manager input, each criterion was weighted, after which an overall score was derived for each location. The result indicates that the municipality intends to prioritize the residential dimension in planning their public charging network which was reflected in the suitability index map. A final step in this study was to perform a cluster analysis to allocate the estimated number of public chargers for each scenario considering the top-scoring locations and wide geographical coverage. This step can also support a socially just energy transition by minimizing the risk of undersupply in weak socio-economic parts of the cities. The final layouts show a concentration of public chargers that exist mainly in residential areas and suburban commercial areas.

In future work, the methodology can be extended in different directions. In estimating the number of charging stations needed, the mobility behavior of one user group was considered and applied for all passenger vehicle owners. An interesting aspect would be to include a variety of vehicle classes and user groups—such as taxis, commuters, public authorities' fleets, etc. In addition, different public charging use cases that exist require different types of chargers which can be considered. The mobility behavior was based on the Mobility in Germany (MiD) latest survey in 2017 which is carried out every five years. It is vital to reconsider the emergence of new mobility behavior following the pandemic.

Moreover, the data generated from current public charging networks in Germany can be leveraged to gain valuable insights into user charging behavior. The demand and supply criteria could also be reassessed based on new findings of future research and surveys that reveal key characteristics of EV owners. Last but not least, the public charging network layouts proposed can be used as the basis for agent-based simulations to determine

the spatio-temporal charging profile, peak power hotspots, and charger utilization. This information could also be used to plan future investments in grid infrastructure.

**Supplementary Materials:** The following supporting information can be downloaded at: https://www.mdpi.com/article/10.3390/wevj13080131/s1, Document S1: Quesitonnaire.

**Author Contributions:** Conceptualization, A.K. and T.-A.F.; methodology, A.K. and T.-A.F. and D.G.; software, A.K.; validation, A.K. and T.-A.F. and D.G.; formal analysis, T.-A.F.; investigation, A.K.; resources, A.K.; data curation, A.K.; writing—original draft preparation, A.K.; writing—review and editing, A.K. and T.-A.F.; visualization, A.K.; supervision, T.-A.F. and D.G. All authors have read and agreed to the published version of the manuscript.

**Funding:** This research received no external funding.

**Data Availability Statement:** Publicly available datasets were analyzed in this study. This data can be found here: openstreetmap.org/copyright. Restrictions apply to the availability of data provided by the Municpality.

**Acknowledgments:** Special thanks goes to Maxim Blankschein from The Institute for Climate Protection, Energy and Mobility (IKEM) for supporting and facilitating this work in cooperation with Bottrop's municipality and acknowledge support by the Open Access Publication Fund of TU Berlin.

**Conflicts of Interest:** The authors declare no conflict of interest.

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
