# Peer review of "Optimizing Public Charging: An Integrated Approach Based on GIS and Multi-Criteria Decision Analysis"

_wevj, doi:10.3390/wevj13080131_

Round 1
Reviewer 1 Report
The paper Optimizing Public Charging: An Integrated Approach Based on GIS and Multi-Criteria Decision Analysis proposes a GIS based methodology to expand the public charging network infrastructure taking into account different criteria and weights. The paper is well written and the methodology could be adopted in other geographical locations however some of the assumptions and results need to be better explained. 1 - the literature review needs to be expanded and compare other type of methodologies such as data driven techniques, i.e. you can compare with https://doi.org/10.3390/wevj11010018 and https://doi.org/10.1109/ACCESS.2020.3004715 2 - The novelty of the approach is not well defined, what are the shortcomings (line 125) that you are overcoming with the proposed methodology? 3 - How the increasing electricity price can affect the development of the proposed scenarios? 4 - How the methodology can ensure the reachability of rural area is the expansion does not consider it? Is reachability a criteria for the expansion? It does not look like it. How the solution can be adapted in this case? 5 - How can you assert that this analysis (line 792) is flexible and scalable if you have tried only on a single region?Author Response
Please see the attachment.

Reviewer 2 Report
This was an excellent paper within the scope of the journal, which I recommend for publication after some minor but necessary adjustments for clarification.
Line 129: Please offer some examples of what the modifications were.
Line 133: Please correct the word “thesis” to “paper”
Line 261-262: Was the attractiveness determined by the stakeholders or the authors? Please clarify.
Line 311:Please remind the reader what the parameters are.
Line 326:Please state the share of the passenger car fleet that is EVs. I see it is in Figure 5, but good to have it also in text.
Line 347: Is this the latest version of this survey? If not, how do the latest survey results have changed?
Line 387: Isn’t there a possibility for flexible tariffs? Please clarify.
There seems to be a problem with the caption of Figure 6.
Line 440-443: Worth commenting that this can be a big barrier for some cases where the DSOs are unwilling to provide this information.
Line 459-460: Please explain why this assumption was made, and provide source, if there is any.
Line 482-483: Worth recommending that municipalities gather data on housing types for future improvements.
Line 502-511: While the rationale is justifiable, it is not equitable. Please comment on equity implications of linking public charging deployment to income and employment data, including previous literature, see below:
DOI: 10.51414/sei2022.020
DOI: 10.1016/j.tranpol.2020.10.003
DOI: 10.38105/spr.e10rdoaoup
Line 581-582: What are the implications of this? Please explain and comment on ways to address this.
Line 610-611: Did you test with other values as well? How would the results change?
Figure 13-16: Could these maps be overlaid with the cluster maps?
Line 778-780: Could you provide examples from the literature regarding utilisation rates?
Line 818: Check section numbering.
Round 2
Reviewer 1 Report
The authors addressed all the comments.